# Expanding the MECP2 network using comparative genomics reveals potential therapeutic targets for Rett syndrome

Irene Unterman[1], Idit Bloch[1], Simona Cazacu[2], Gila Kazimirsky[3], Bruria Ben-Zeev[4], Benjamin P Berman[1]*, Chaya Brodie[3]*, Yuval Tabach[1]*

[1]Department of Developmental Biology and Cancer Research, Institute for Medical Research Israel-Canada, Jerusalem, Israel; [2]Hermelin Brain Tumor Center, Henry Ford Hospital, Detroit, United States; [3]The Mina and Everard Goodman Faculty of Life Sciences, Bar-Ilan University, Ramat-Gan, Israel; [4]Edmond and Lily Safra Children's Hospital, Chaim Sheba Medical Center, Ramat Gan, Israel

**Abstract** Inactivating mutations in the Methyl-CpG Binding Protein 2 (MECP2) gene are the main cause of Rett syndrome (RTT). Despite extensive research into MECP2 function, no treatments for RTT are currently available. Here, we used an evolutionary genomics approach to construct an unbiased MECP2 gene network, using 1028 eukaryotic genomes to prioritize proteins with strong co-evolutionary signatures with MECP2. Focusing on proteins targeted by FDA-approved drugs led to three promising targets, two of which were previously linked to MECP2 function (IRAK, KEAP1) and one that was not (EPOR). The drugs targeting these three proteins (Pacritinib, DMF, and EPO) were able to rescue different phenotypes of MECP2 inactivation in cultured human neural cell types, and appeared to converge on Nuclear Factor Kappa B (NF-κB) signaling in inflammation. This study highlights the potential of comparative genomics to accelerate drug discovery, and yields potential new avenues for the treatment of RTT.

*For correspondence:
ben.berman@mail.huji.ac.il (BPB);
chaya@brodienet.com (CB);
yuvaltab@ekmd.huji.ac.il (YT)

**Competing interests:** The authors declare that no competing interests exist.

## Introduction

Rett syndrome (RTT [MIM: #312750]) is a rare genetic disorder caused by mutations in the methyl-CpG binding protein 2 (*MECP2*) gene (*Amir et al., 1999*). It is an X-linked dominant disorder, almost exclusively affecting females. RTT is characterized by normal early development followed by regression, loss of purposeful hand movements, and intellectual disability (*Liyanage and Rastegar, 2014*). Characteristics of the disease such as age of onset, range of symptoms, and their severity vary between patients (*Neul et al., 2008*). The MECP2 protein has a role in transcriptional regulation but its exact mechanism of action, co-factors and downstream targets, are not fully characterized (*Picard and Fagiolini, 2019*).

The MECP2 protein, as a transcriptional regulator, has been linked to both gene silencing and activation (*Horvath and Monteggia, 2018*). It binds methylated CG dinucleotides through its methyl-binding domain and has a transcriptional repression domain that mediates binding to co-repressor proteins (*Lyst et al., 2013*). These two domains are hotspots for missense, nonsense, and frameshift mutations for RTT disease (*Ragione et al., 2016*; *Krishnaraj et al., 2017*). Transcriptomic analyses of RTT patient postmortem brain tissues, blood samples, cell lines, and murine models (*Shovlin and Tropea, 2018*) have revealed hundreds of affected genes but with a low degree of overlap between studies (*Picard and Fagiolini, 2019*).

Therapies aiming to restore *MECP2* loss of function are in early development stages and are not approved for patients. These include attempts at read-through inducing drugs to overcome nonsense mutations (*Brendel et al., 2011*; *Merritt et al., 2020*), X reactivating therapies to allow

expression of silenced endogenous *MECP2 (12)*, and gene therapy treatments to introduce wild-type *MECP2* copies into the brain (*Guy et al., 2007*; *Garg et al., 2013*). Furthermore, current gene therapy methods show limited transduction efficiency and are unable to modify all cells in the target tissue (*Gadalla et al., 2013*). Targeting factors downstream of MECP2 offers a promising avenue which could allow for partial amelioration of Rett symptoms, without the severe neurological phenotypes associates with MECP2 overexpression (*Chao and Zoghbi, 2012*; *Ramocki et al., 2009*).

One major downstream target of MECP2 is brain-derived neurotrophic factor (BDNF) (*Chen et al., 2003*), a key modulator of neuronal development and function. BDNF levels are reduced in symptomatic MECP2 KO mice (*Kline et al., 2010*) and experimental interventions that increase BDNF levels improve RTT-like phenotypes (*Chang et al., 2006*). The IGF-1 protein potentiates BDNF activity, and treatments with recombinant human IGF-1 and its analogs have led to improvement in several clinical measurements (*Khwaja et al., 2014*; *Glaze et al., 2019*), with a trial of an IGF-1 synthetic analog currently in Phase 3 (NCT04181723). Additional trials include the administration of triheptanoin supplementation which is being investigated in a Phase two clinical trial (NCT02696044) to target mitochondrial dysfunction in RTT (*Park et al., 2014*). Other interventions, such as ketamine (NCT03633058), targeting N-methyl-D-aspartate receptor (NMDAR) dysfunction, and Anavex 2–73 (NCT03758924), aimed at restoring mitochondrial dysfunction (*Kaufmann et al., 2019*), are currently being studied. Histone deacetylases (HDACs) mediate MECP2 gene repression, and have also been suggested as therapeutic targets for Rett syndrome (*Shukla and Tekwani, 2020*), which may act on BDNF trafficking (*Xu et al., 2014*). The lack of a successful therapy to date emphasizes the need for accurate and comprehensive mapping of the MECP2 network with a focus on targetable proteins that could be used for interventions (*Vashi and Justice, 2019*). Here, we use a genomics approach that has thus far not been applied to MECP2 (phylogenetic profiling), combined with data from drug target databases, to construct such a map.

A well-tested approach for gene functional prediction and identification of functional gene networks is phylogenetic profiling (PP), which analyzes gene conservation patterns across an evolutionary related set of organisms (*Pellegrini et al., 1999*). Genes with similar evolutionary patterns are functionally coupled, irrespective of sequence homology (*Date and Marcotte, 2003*). PP was used successfully to predict protein function (*Enault et al., 2004*; *Eisen and Wu, 2002*; *Jiang, 2008*; *Merchant et al., 2007*), protein interactions (*Kim and Subramaniam, 2006*; *Sun et al., 2005*) and protein localization within the cell (*Marcotte et al., 2000*; *Pagliarini et al., 2008*; *Avidor-Reiss et al., 2004*; *Hodges et al., 2012*). When a pathway becomes non-functional in a specific lineage (often due to loss of a key gene in the pathway), then other genes involved in the pathway may lose the fitness advantage they provide to the organism if they are not involved in other important pathways/functions. PP captures these pathway-level loss events, providing a quantitative measurement of functional relatedness and prioritizing those genes minimally involved in the same pathway over pleiotropic genes required in multiple pathways. This property is especially attractive for drug development, where the goal to is target the most specific proteins possible. Because PP is based on evolutionary dynamics rather than protein expression or sequence/structure features, it produces results that are highly complementary to traditional methods (*Dey et al., 2015*; *Arkadir et al., 2019*).

In our recent work, we developed an improved version of PP that normalizes signals across hundreds of eukaryotic genomes to identify important new members of cellular and disease pathways (*Arkadir et al., 2019*; *Sherill-Rofe et al., 2019*; *Tabach et al., 2013a*; *Tabach et al., 2013b*; *Schwartz et al., 2013*). As protein function and evolution are very complex (*Bloch et al., 2020*), co-evolution within a local lineage (i.e. mammals, vertebrates, fungi, etc.) can provide complementary evidence, and is especially informative for genes with multiple functions that show complex or recent evolution (*Sherill-Rofe et al., 2019*). We recently incorporated a clade-specific analysis into our Normalized Phylogenetic Profile (NPP) software pipeline (*Tsaban et al., 2021*), which is used for all the analyses in this study. We have used several functional databases including CORUM (*Ruepp et al., 2008*), KEGG (*Kanehisa and Goto, 2000*), and REACTOME (*Jassal et al., 2020*) to validate the NPP method and show that it recovers known protein complexes and pathways with higher specificity and sensitivity than other phylogenetic methods (*Bloch et al., 2020*; *Tsaban et al., 2021*).

We combined our PP-derived interaction map of MECP2 with drug targeting databases to select new candidate drugs for RTT. There are thousands of known bioactive compounds collected in databases such as Drug-Gene interaction database (DGIdb) (*Cotto et al., 2018*) and Open Targets

(*Oxford Academic, 2019*), that contain information about validated direct or indirect effect on specific proteins or pathways. Network-based drug discovery methods are emerging as important tools to identify novel drug targets and predict the drug's mode of action and potential side-effects (*Iorio et al., 2013*), and have been associated with higher success rates in clinical trials (*Nelson et al., 2015*; *Pushpakom et al., 2019*). From our PP network, we selected three proteins that showed robust co-evolution with MECP2 across multiple evolutionary scales, had related roles in inflammation, and could be targeted by pharmacologically-validated compounds. These three compounds – pacritinib, EPO and DMF – were then functionally validated in *MECP2* Knock-Down (KD) human neural cells, including microglia, astrocytes and neural stem cells. The ability of all these drugs to reverse some aspects of the MECP2-KD phenotypes demonstrates how comparative genome analysis can be used to understand and target this disease network.

## Results

### Phylogenetic profile analysis of MECP2 in eukaryotes and mammals identifies MECP2 co-evolved genes

To trace the evolutionary history of *MECP2*, we used BLASTP (*Camacho et al., 2009*) to compare the MECP2 protein to the proteomes of 1028 eukaryotic species. We repeated the process for each of 20,192 human transcripts to generate a profile of conservation for each human gene. The scores were normalized according to sequence length and the evolutionary distance between humans and the queried species as described previously (*Tabach et al., 2013b*; *Bloch et al., 2020*). This Normalized Phylogenetic Profile (NPP) describes how conserved a transcript is in each species across the tree of life, compared to its expected conservation. Genes with similar NPPs (i.e. genes that are either conserved or lost as a group) are functionally related, often belonging to the same pathway (*Sherill-Rofe et al., 2019*; *Tabach et al., 2013a*; *Tabach et al., 2013b*; *Schwartz et al., 2013*). We have shown that both across-clade and within-clade measures of co-evolution can be informative, and thus we computed NPPs both across all eukaryotes (*Figure 1A* and *Figure 1—figure supplement 1A*) and within mammals (*Figure 1B* and *Figure 1—figure supplement 1B*) as described previously (*Sherill-Rofe et al., 2019*; *Bloch et al., 2020*; *Tsaban et al., 2021*; *Braun et al., 2020*).

We first used NPP scores to sort all human genes by their phylogenetic similarity to MECP2 across eukaryotes (*Figure 1A* and *Figure 1—figure supplement 1A*). We selected the top 200 genes (the 'E200' set), which showed high correlation (R>0.47) with *MECP2*. Similarly, we used NPP scores within the mammalian clade (*Figure 1B* and *Figure 1—figure supplement 1B*) to select the top 200 mammal-centric genes (the 'M200' set), which also showed high *MECP2* correlations (R>0.49) (*Supplementary file 1*) [code is available on GitHub (*Unterman et al., 2021*; copy archived at swh:1:rev:14063b44a40688a8024a06347b63cfdac74b96ad)]. These two lists find correlated evolution at different scales, billions of years for eukaryotes and hundreds of millions for mammals. Therefore, the M200 set may show genes more recently co-evolved with MECP2. E200 and M200 had very different functional properties, based on STRING enrichment to Gene Ontology and other annotation sets (*Supplementary file 2*) The E200 list was enriched in a number of different functional categories, with some of the most significant related to innate immunity and the immune response. These included MHC Class II receptor activity and peptidoglycan receptor activity in GO, and the associated autoimmunity Asthma, Allograft rejection, and Type I in KEGG. In contrast, the M200 list had almost no functional enrichment, except for the general 'Disease' category within Uniprot annotated keywords.

The E200 and M200 lists had 10 genes in common (*Figure 1C*). As mammals are included within the eukaryote analysis, the two lists are not independent and thus the expected degree of overlap is hard to determine. Nevertheless, in two independent gene lists of this size we would only expect two overlapping genes. We thus considered the set of 10 overlapping genes, as well as the full union set of 390 genes, as two tiers of candidates for further exploration. A GeneAnalytics query of the 390 gene union set revealed it was enriched with genes expressed in the brain (p-value < 0.00024), with 79 genes linked to the cerebral cortex and 54 to the cerebellum (*Figure 1—figure supplement 1C*; *Ben-Ari Fuchs et al., 2016*). Only 24 of the 390 gene list were overlapping with a length-matched list of MECP2 interactions from STRING (*Figure 1—figure supplement 1D,E*), highlighting

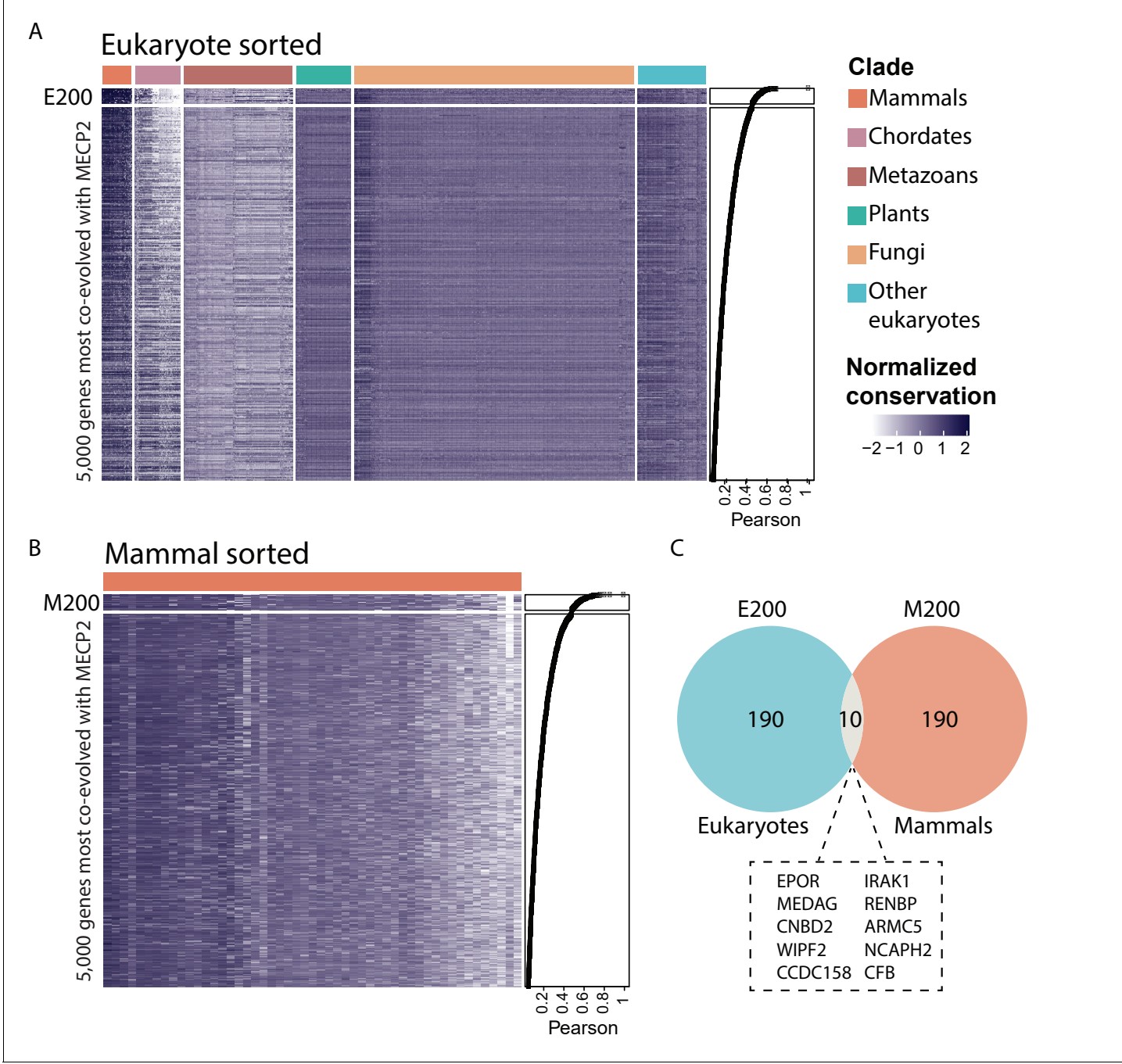

**Figure 1.** The phylogenetic profile of MECP2. (**A**) The Normalized Phylogenetic Profile (NPP) of *MECP2* in eukaryotes along with the top 5000 co-evolved proteins. Each row represents one human protein, ordered by Pearson correlation to *MECP2,* with the top 200 proteins labeled as E200. Darker color indicates higher normalized conservation of the protein in each species. Organisms are grouped by phylogenetic clade, and clustered within each clade. (**B**) The NPP of *MECP2* in 51 mammals along with the 5000 top co-evolved proteins, ordered by Pearson correlation to MECP within mammals. The top 200 proteins are labeled as M200. (**C**) Gene overlap of the top 200 most co-evolved genes in eukaryotes and mammals, listing the 10 gene names listed in both.

The online version of this article includes the following figure supplement(s) for figure 1:

**Figure supplement 1.** Scaled phylogenetic profiles of all human genes by similarity to MECP2.

the complementary role phylogenetic analysis to other measurements and predictions of protein interaction.

## MECP2 druggable protein network identifies several drug targets located in evolutionarily conserved chromosomal clusters

We chose to focus on the subset of *MECP2* co-evolved genes that could be targeted by readily available compounds. This approach allows for straightforward functional testing in both in vitro and animal models of *MECP2* inactivation and moves us closer to the ultimate goal of identifying a candidate therapeutic for RTT. Using DGIdb (*Cotto et al., 2018*) and Open Targets (*Oxford Academic, 2019*), we searched specific drug-gene interaction types (e.g. inhibitory, activating) to identify direct effects on the 390 candidate genes. Open Targets identified such protein-compound interactions for 11 proteins from our list, and DGIdb identified an additional 22, resulting in 33 druggable proteins in the MECP2 phylogenetic network. MECP2-linked proteins were neither more or less druggable than other proteins in the genome, since the number expected by chance was 39.9±6.0.

Two of the 33 druggable NPP-linked proteins (IRAK1 and EPOR) were found in both the eukaryotic E200 and the mammalian M200 lists, and were targeted by compounds with clinical efficacy and acceptable safety profiles (Pacritinib and EPI), making these immediately attractive for further study. In a STRING (*Szklarczyk et al., 2019*) functional interaction map, which is based on independent data types, IRAK1 was one of the six proteins functionally linked to MECP2, whereas EPOR had no known functional links (*Figure 2A*). To identify additional targets among the 31 druggable NPP-linked proteins found in only one of the E200 and M200 lists, we investigated chromosomal clusters of genes with MECP2-linked genes, since clustered genes often indicate a shared evolutionary function. Indeed, IRAK1 is directly adjacent to MECP2 on an evolutionary conserved chromosomal band on the X chromosome (*Figure 2B* and *Figure 2—figure supplement 1*). In a chromosomal clustering analysis of all MECP2-linked genes, the chr19p13.2 band contained more MECP2 co-evolved genes than any other in the genome (*Figure 2C*). Unexpectedly, this 5.6 Mb domain contained 5 of the 33 druggable MECP2 coevolved genes - EPOR, DNMT1, ICAM1, ICAM3, and KEAP1 (*Figure 2D*), making this the most strongly enriched band in the genome (p<1E-5, *Figure 2D* and *Figure 2—figure supplement 2*). Topologically Associated Domains (TADs) define the 3D organization of functionally co-regulated gene clusters (*Fritz et al., 2019*), and the entire chr19p13.2 gene cluster defines a single TAD-based high-resolution Hi-C mapping (*Rao et al., 2014*; *Figure 2E*). Additionally, this region has been shown to be bound by MECP2 (*Yasui et al., 2007*).

For the two druggable proteins found in both NPP lists (IRAK1, EPO1) and the other four proteins located in the chr19p13.2 TAD (DNMT1, ICAM1, ICAM3, KEAP1), we further investigated the associated drug's mode of action and efficacy and safety profile in other contexts. IRAK1 has been identified as a key intermediary in NF-κB signaling in RTT, with MECP2 directly binding the IRAK1 promoter to regulate its expression in MECP2-null mouse brains and *NFKB1* dramatically increasing lifespan in *MECP2* null mice (*Kishi et al., 2016*). Pacritinib is a JAK2/FLT3 inhibitor with IRAK1 inhibiting capabilities (*Jensen et al., 2017*), presently under clinical investigation for myelofibrosis and glioblastoma. As it is currently the only clinical stage IRAK1 inhibitor with known clinical efficacy and acceptable safety even after prolonged administration (*Singer et al., 2018*), we chose this compound for our functional studies to target IRAK1.

EPOR forms a heterodimer with β common receptor (βCR), also known as the Tissue-Protective Receptor (TPR) (*Brines et al., 2004*). The endogenous hormone EPO binds the TPR activating signaling cascades with roles in neuroprotection (*Rey et al., 2019*). EPO has a very well understood pharmacological profile, and several variants of erythropoietin lacking hematopoietic effects have been developed and shown to preserve the neuroprotective effects of EPO administration (*Chen et al., 2015*; *Patel et al., 2011*).

KEAP1 is the key repressor of the transcription factor nuclear factor erythroid 2-related factor 2 (NFE2L2), which is involved in response to antioxidants through its transcriptional regulation of inflammatory pathways. Dimethyl fumarate (DMF) is an approved treatment for both multiple sclerosis and psoriasis (*Torkildsen et al., 2016*; *Blair, 2018*). Its administration decreases KEAP1 levels, increases NFE2L2 levels and causes dissociation and nuclear transport of NFE2L2. DMF treatment has proven to be effective and safe in conferring neuroprotection (*Montes Diaz et al., 2018*).

Since the three candidate proteins targeted by these drug products (IRAK1, EPOR, and KEAP1) all have possible roles mediating inflammation, we chose to proceed with these for functional studies

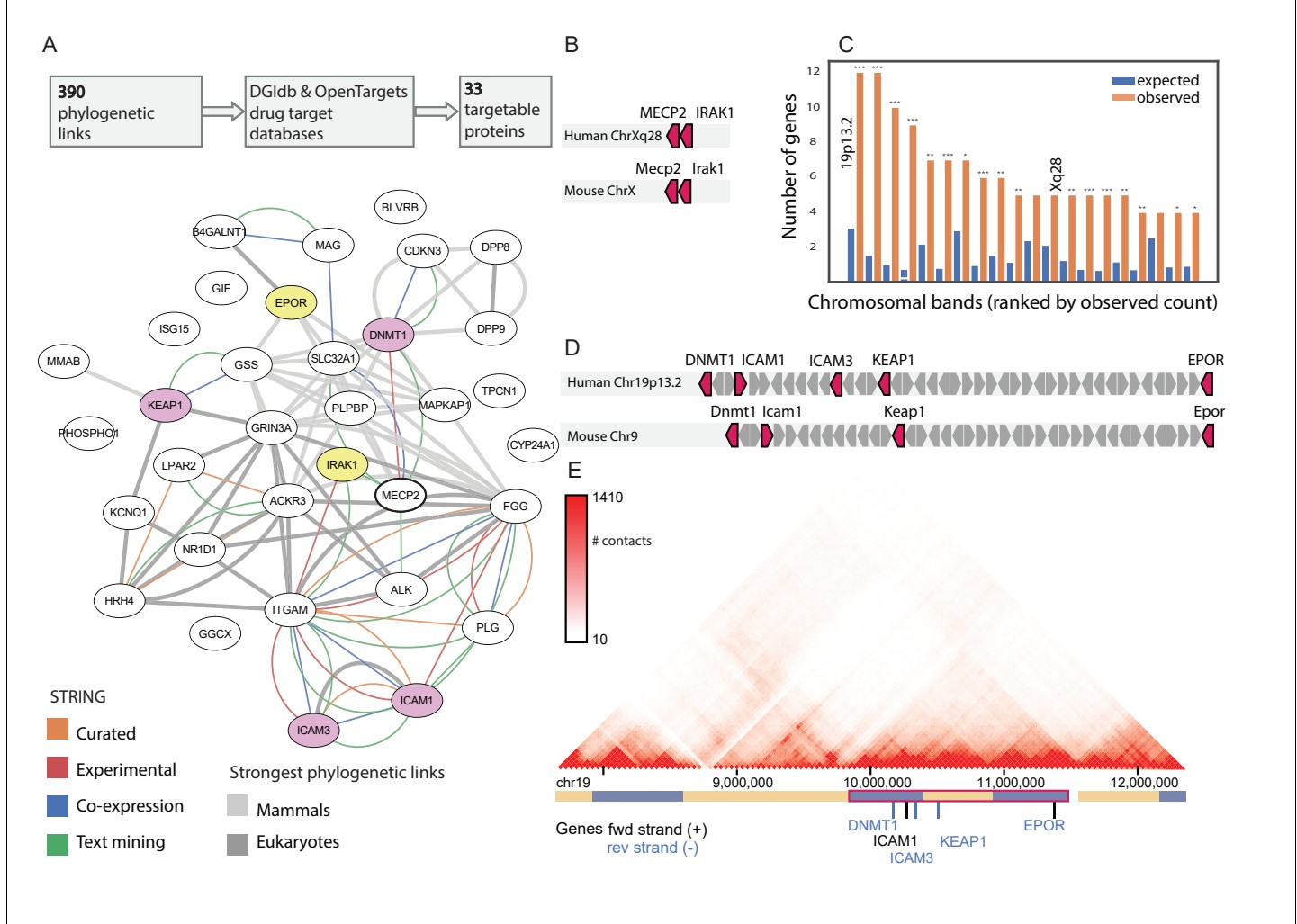

**Figure 2.** The *MECP2* druggable genes network. (**A**) The 390 genes co-evolved with *MECP2* in either eukaryotes or mammals contain 33 genes that are targets of known drugs/compounds. These genes are shown in a STRING interaction graph, with known gene-gene interactions shown as colored edges, and those with very strong co-evolution (Pearson correlation > 0.7) shown as gray edges. The two proteins co-evolved in both eukaryotes and mammals (EPOR and IRAK1) are colored yellow and the four other genes found at chr19p13.2 are colored purple. (**B**) Genomic location of MECP2 interacting genes along chromosome X in humans and mice. (**C**) Karyotype band locations of MECP2 and the 390 co-evolved genes. * p<0.05, **p<0.01, *** p<0.001 (**D**) Genomic location of MECP2 interacting genes along chr19p13.2 in humans and chromosome nine in mice. (**E**) Intra-chromosomal Hi-C contact heatmap for the chr19p13.2 locus in GM12878 cells, adapted from the 3D Genome Browser (*Wang et al., 2018*). Genes in the MECP2 network are shown, with other genes hidden for clarity. TADs called by 3D Genome Browser shown in blue/yellow track, with a super-TAD containing MECP2 genes outlined in red.

The online version of this article includes the following figure supplement(s) for figure 2:

**Figure supplement 1.** Synteny in the Chr19p13.2 locus.

**Figure supplement 2.** Synteny between MECP2 and IRAK1.

in cells modeling neural inflammatory phenotypes. The other three proteins identified (DNMT1, ICAM1, and ICAM3) may also be interesting targets for future study.

## Human neural cell cultures exhibit Rett-like phenotypes when MECP2 is silenced

To analyze the effects of the three selected compounds on a RTT model system, we employed human immortalized primary neural cells silenced for MECP2. We chose to use a human in vitro model system for preliminary validation due to the significant differences in the phenotypes and functions of mouse and human neural cells (*Xu et al., 2018*). Indeed, recent studies reported the use

of iPSCs and immortalized glial cells as reliable human models for analyzing specific therapeutic targets for various diseases including autism (*Tsilioni et al., 2020*), neuroinflammation (*Chiavari et al., 2019*; *Timmerman et al., 2018*) and for analyzing the crosstalk of glioma with glial cells (*Henrik Heiland et al., 2019*; *Zeng et al., 2020*; *Bier et al., 2020*).

Since all three selected compounds were associated with changes in neuroinflammation, we first focused on microglia and astrocytes. Loss or impaired function of MECP2 has been reported to affect the functions of glial cells and these changes have been implicated in the pathogenesis of RTT (*Kahanovitch et al., 2019*; *Jin et al., 2017*). We first characterized the effects of MECP2 silencing on the phenotypes of microglia (*Figure 3*). For these experiments, we transduced the cells with lentivirus vectors expressing control or MECP2 shRNAs. The expression of MECP2 was silenced by over 80% in these cells (*Figure 3—figure supplement 1*). MECP2 silencing induced a relative increase in the expression of M1 markers (IL1, and CD86) and a relative decrease of M2 markers (CD206 and IL-13) (*Figure 3A*), suggesting an increase in neuroinflammation. We also found that MECP2 silencing decreased the phagocytosis of the microglia cells (*Figure 3B*).

We next analyzed the effect of MECP2 silencing on human astrocyte differentiation and functions. For these experiments, neural stem cells (NSCs) were transduced with lentivirus vectors expressing the differentiation reporters, GFP/fLuc GFAP or the mCherry/Luc MAP2. The cells were plated on laminin coated plates and luciferase activity was determined 10 days later. In accordance with previous reports (*Andoh-Noda et al., 2015*; *Mok et al., 2020*), we found that MECP2 silenced NSCs displayed preferential expression of the glial differentiation marker (GFAP) with a concomitant decrease in expression of the neuronal differentiation marker (MAP2) (*Figure 3C*). The MECP2 silenced astrocytes expressed lower levels of Excitatory Amino Acid Transporter 2 (EAAT2), indicating impaired function of these cells, as was suggested for mouse astrocytes (*Okabe et al., 2012*; *Figure 3D*).

We finally examined the effects of MECP2 silencing on the expression of BDNF in human neurons and astrocytes and found that, as reported for mouse cells (*Chang et al., 2006*; *Li et al., 2012*), MECP2 silencing decreased the expression of BDNF mRNA in both astrocyte and neuronal cultures (*Figure 3E*). Altogether, these finding indicate that the immortalized human cells employed in this study represent a reliable model for studying MECP2-related pathways and potential treatments.

## DMF, EPO, and pacritinib differentially abrogate the effects of MECP2 silencing on microglia polarization

We then analyzed the effects of the three selected compounds, DMF, EPO and pacritinib, on MECP2 silenced cells. We first demonstrated that in the concentrations range we used, none of these compounds exerted a toxic effect on neither microglia cells, astrocytes, primary neuronal cells nor NSCs (*Figure 3—figure supplement 2*).

Control and MECP2 silenced microglia were treated with the DMF, EPO, and pacritinib, and their effects on the relative expression M1 (IL1, TNFα and CD86) and M2 (CD206 and IL-13) markers were determined using RT-PCR. While MECP2 wild-type cells were unchanged by the three compounds, all three abrogated the M1 shift induced by MECP2 silencing, albeit to a different degree (*Figure 4A–C*). EPO exerted the most significant effect, abrogating the effects of MECP2 silencing in all five marker genes (*Figure 4A*). In contrast, both DMF and pacritinib exerted only partial effects, abrogating the effects of MECP2 silencing in CD206, CD86, and IL-1, but failing to abrogate the effects on IL-13 and TNF-α (*Figure 4B–C*). We also analyzed the effects on phagocytosis in microglia cells. Treatment of MECP2 silenced cells with DMF exerted a significant increase in the phagocytosis of the silenced cells, whereas EPO and pacritinib did not (*Figure 4D*).

## EPO and DMF inhibit NF-κB activation in MECP2 silenced microglia cells

Knockdown of MECP2 has been reported to lead to increased NF-κB activity in neuronal and myeloid cells (*O'Driscoll et al., 2015*) and this deregulated activity has been associated with neuroinflammation and decreased dendritic arborization and spine density in a mouse model of RTT (*Kishi et al., 2016*). We therefore examined the ability of our three candidate compounds to abrogate the increased NF-κB activity in MECP2 KD microglia cells. MECP2-silenced microglia cells were transfected with lentivirus vectors expressing the NF-κB luciferase reporter and a constitutively active Renilla luciferase construct (as a transfection control). The cells were treated with EPO, DMF

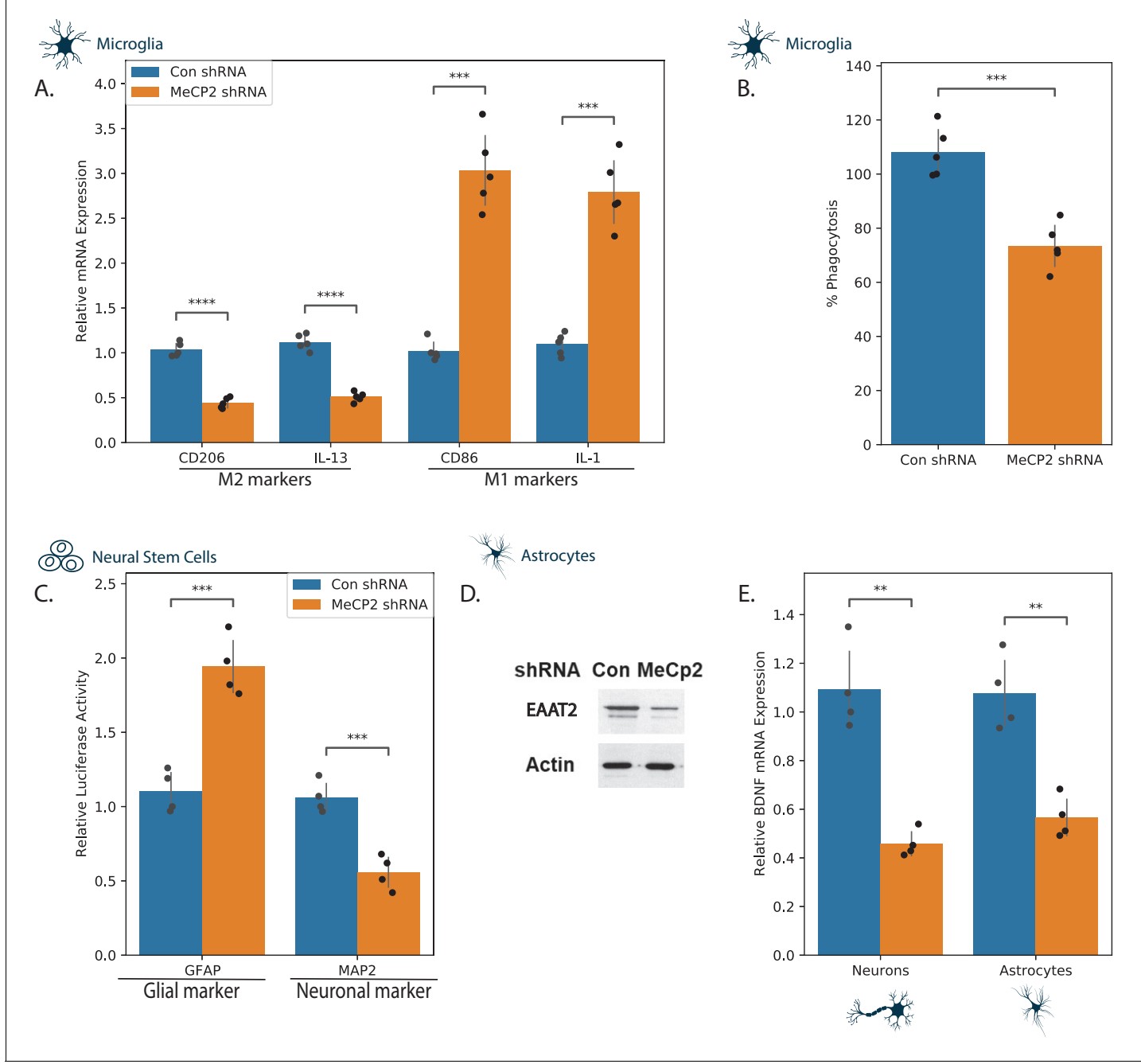

**Figure 3.** Effects of MECP2 knockdown on neural cell phenotypes. Human microglia (**A,B**), NSCs (**C**), astrocytes (**D,E**), and neurons (**E**) were silenced for MECP2 using lentivirus vector. The relative expression of M1 and M2 markers was analyzed in microglia cells using RT-PCR (**A**) and degree of phagocytosis using the pHrodo assay (**B**). NSCs were transduced with lentivirus vectors expressing the differentiation reporters GFAP and MAP2 and were differentiated as described in the methods. Ten days later, luciferase activity was determined (**C**). Astrocytes silenced for MECP2 were analyzed for the expression of EAAT2 using western blot analysis (**D**) and the expression of BDNF mRNA in both MECP2 silenced astrocytes and neurons was determined using RT-PCR (**E**). The results are a representative experiment of three separate tests analyzed in quadruplet. **p<0.01, ***p<0.001, ****p<0.0001.

The online version of this article includes the following figure supplement(s) for figure 3:

**Figure supplement 1.** Silencing of MECP2 in neural cells.

**Figure supplement 2.** Cytotoxic effects of tested compounds.

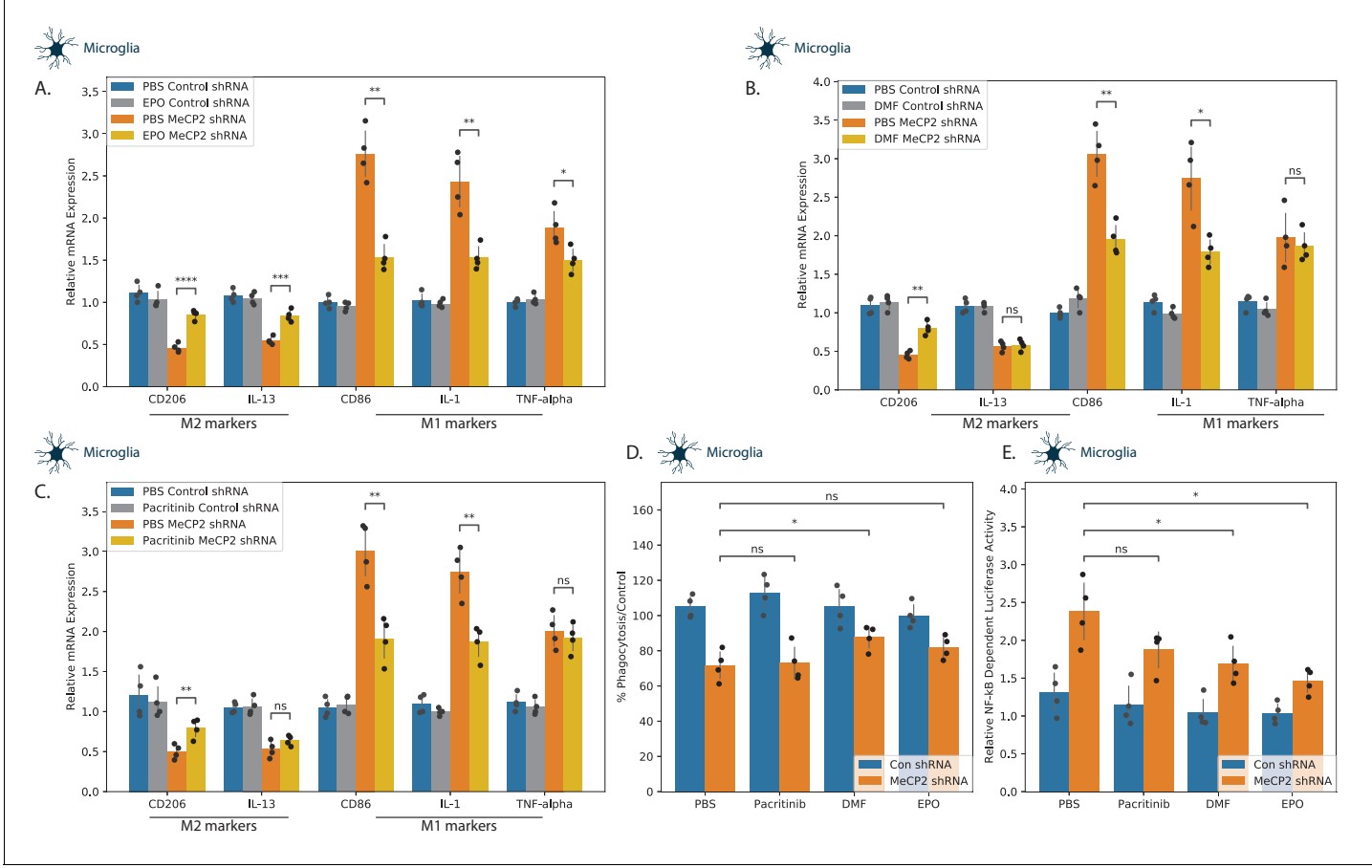

**Figure 4.** Effects of tested compounds on the polarization of microglial cells. Human microglia cells were silenced for MECP2 using lentivirus vectors expressing MECP2 shRNA. Control cells were transduced with lentivirus vectors expressing a control shRNA. After 5 days, the expression of MECP2 was determined by western blot analysis (*Figure 3—figure supplement 1*). Control and silenced cells were treated with EPO 10 ng/ml (**A**), DMF 10 µM (**B**) and pacritinib 10 µM (**C**), and the expression of M1 and M2-associated markers were determined after 72 hr using RT-PCR. MECP2 silenced microglia cells treated with EPO, DMF, or pacritinib were also analyzed for phagocytosis using the pHrodo assay (**D**). MECP2-silenced microglia cells were transduced with lentivirus vectors expressing the NF-kB reporter followed by treatment with EPO, DMF, and pacritinib for 24 hr. Luciferase activity was determined (**E**). The results demonstrate the means ± SD of a representative experiment of three separate tests analyzed in quadruplet. *p<0.05, **p<0.01, ***p<0.001,****p<0.0001.

and pacritinib for 24 hr and luciferase activity was determined thereafter. A significant increase in NF-κB-dependent luciferase activation was observed for EPO and DMF in the MECP2-silenced cells (*Figure 4E*), indicating that EPO and DMF were able to downregulate the increased NF-κB activation in MECP2 KD cells.

## EPO, but not DMF or pacritinib, abrogate improper astrocytic differentiation and astrocyte function in MECP2-silenced neural cells

Various studies support a non-cell-autonomous effect of astrocytes on neuronal cell functions and contribution of astrocytes to various aspects of Rett syndrome pathogenesis via regulation of glutamate levels, homeostasis, and neuroinflammation (*Kahanovitch et al., 2019*; *Maezawa et al., 2009*). As presented in *Figure 3C*, MECP2 silencing increased astrocytic differentiation of NSCs at the expense of neuronal differentiation. Treatment of the silenced NSCs with pacritinib or DMF did not have significant effects on markers of neuronal (βIII tubulin) or astrocytic (GFAP) differentiation (*Figure 5A*). In contrast, EPO abrogated the impaired NSC differentiation as reflected by an increase of βIII tubulin and a decrease of GFAP (*Figure 5A*). In human astrocytes, MECP2 silencing decreased the expression of the glutamate transporter EAAT2 (*Figure 3D*). Treatment of these cells with EPO

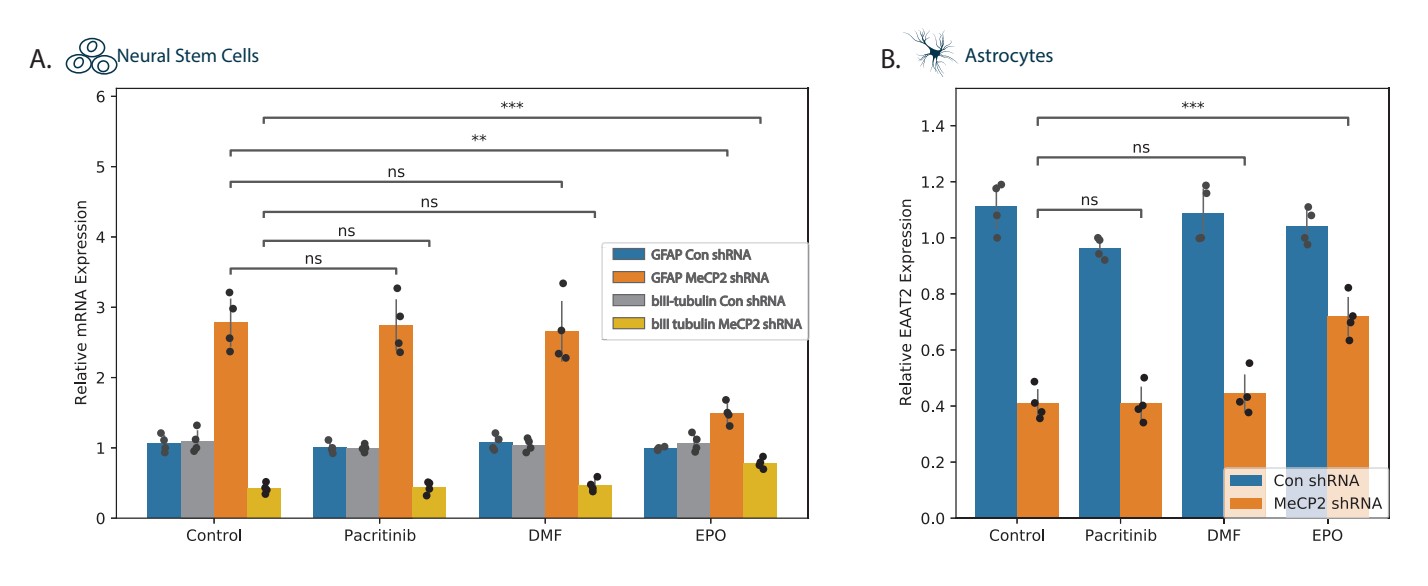

**Figure 5.** Effects of tested compounds on the differentiation of NSCs and EAAT2 expression in astrocytes. Human NSCs (**A**) and astrocytes (**B**) were silenced for MECP2 using lentivirus vectors expressing MECP2 shRNA. Control cells were transduced with lentivirus vectors expressing a control shRNA. After 5 days, the expression of MECP2 was determined by western blot analysis (*Figure 3—figure suppplement 1*). (**A**) Control and silenced NSCs were allowed to differentiate for 10 days and the expression of differentiation markers GFAP and βIII tubulin were determined using RT-PCR. (**B**) Silenced astrocytes treated with the different compounds or with medium were analyzed for the expression of EAAT2 using RT-PCR. The results are of a representative experiment of three separate tests analyzed in quadruplet. *p<0.05, **p<0.01, ***p<0.001,****p<0.0001.

inhibited the decreased EAAT2 expression, whereas pacritinib and DMF did not have a significant effect (*Figure 5B*).

## DMF and EPO upregulate BDNF expression in MECP2-silenced cells

BDNF plays important roles in neuronal growth and development and represents a well-recognized transcriptional target of MECP2 (*Chen et al., 2003*). Silencing of *MECP2* decreased BDNF

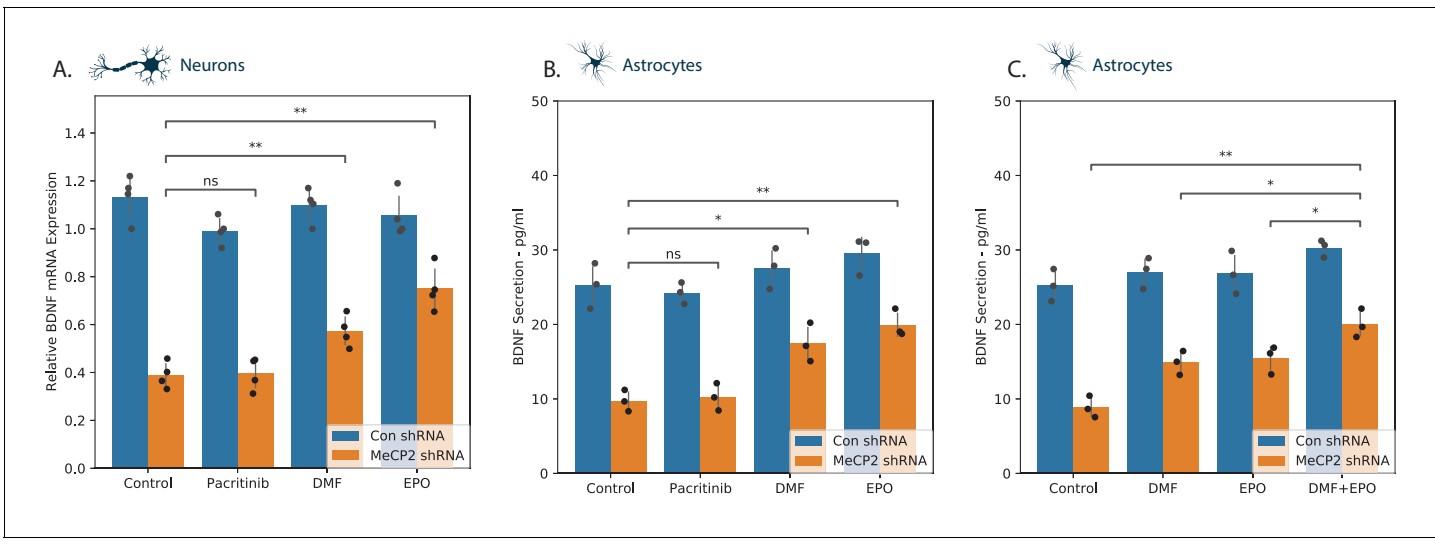

**Figure 6.** BDNF expression in neural cells. MECP2 expression was silenced in human neurons (**A**) and astrocytes (**B, C**) and BDNF expression was analyzed in cells treated with a medium or the specific tested compounds using both RT-PCR (**A**) and ELISA (**B, C**). The combinatorial effect of EPO and DMF was assessed on BDNF expression in astrocytes (**C**). *p<0.05, **p<0.01, ***p<0.001,****p<0.0001.

expression in both neurons and astrocytes (*Figure 6A–B*). DMF and EPO induced an increase in BDNF mRNA in the silenced neurons (*Figure 6A*) and in the silenced astrocytes (*Figure 6B*), whereas pacritinib did not exert a significant effect in either. We replicated this experiment in astrocytes, testing DMF and EPI both individually and in combination (*Figure 6C*). This showed a modest increase in the level of BDNF in the in the combined treatment, suggesting potentially independent modes of action.

## Discussion

Rett syndrome is a neurodevelopmental disease without evidence of degeneration and thus rescuing MECP2 downstream activity might improve disease pathophysiology even in adulthood (*Faundez et al., 2019*). In murine models, reintroduction of *MECP2* to adult animals led to vast phenotypic improvements (*Guy et al., 2007*). In parallel with approaches aimed at increasing MECP2 levels directly (*Brendel et al., 2011*; *Merritt et al., 2020*; *Przanowski et al., 2018*), accurate mapping of the MECP2 gene network is important to provide additional avenues for intervention, and avoid the toxicity associated with overexpression of MECP2 (*Ramocki et al., 2009*). Here, we have presented a framework for identifying potential drug-gene interactions based on comparative genomics, and validate these interactions in human immortalized primary neural cells. We analyzed phylogenetic profiling across mammals and hundreds of eukaryotes to infer functional links and generated a network of co-evolved genes, identifying known targets such as IRAK1 as well as new candidates such as EPOR. We then mined this network to identify actionable drug targets, prioritizing a list of associated compounds using a combination of strong co-evolutionary evidence of the target proteins and a proven safety profile of approved pharmaceutical compounds (Pacritinib, DMF, and EPO). While we prioritized based on the potential for drug repurposing, our evolutionary analysis could also be used to identify additional targets for drug design. Furthermore, we only performed functional validation 3 of the 33 co-evolved genes associated with approved drugs, leaving others for potential future study.

We reproduced a number of RTT phenotypes by reducing expression of MECP2 in human neural cell cultures. When treated with the three selected compounds, these phenotypes could all be reversed to different degrees in a cell-dependent manner. While all the compounds restored the polarization of microglia and reduced the hyperactivity of NF-κB signaling in microglia, only DMF abrogated decreased phagocytosis. EPO alone reversed the impaired neuronal/astrocyte differentiation of NSCs. In neurons and astrocytes, DMF and EPO abrogated the inhibitory effect of *MECP2* silencing on BDNF expression and secretion, whereas pacritinib did not have a significant effect.

The activity of these compounds on microglia polarization and NF-κB activity is mechanistically interesting. Activation of microglia cells has been reported in *MECP2* KO mice (*Zhao et al., 2017*) and has been associated with pathological process in RTT (*Kahanovitch et al., 2019*). An MECP2 KO mouse model also implicated NF-κB signaling and IRAK1 as important mediators of the neuroinflammatory response in RTT (*Kishi et al., 2016*). While the mechanisms involved in the reversing effects of these three compounds remains to be determined, NF-κB signaling downstream of MECP2 is an attractive candidate (*Figure 7*). EPOR and KEAP1 have also been implicated as NF-κB modulators involved in the inflammatory response. The endogenous hormone EPO binds the TPR to suppresses inflammatory cytokines in immune cells, a process which involves inhibition of the NF-κB pathway through GSK3β (*Patel et al., 2011*). NFE2L2, usually bound by KEAP1, also has a direct cytoplasmic role in modulating NF-κB signaling through degradation of IκB and nuclear translocation of NF-κB (*Ganesh Yerra et al., 2013*). The three compounds tested all inhibited NF-κB-dependent luciferase activation in microglia cells to a certain extent, validating their involvement in this context.

In addition to the inflammatory effect, we found that *MECP2* silencing decreased the neuronal while increasing the astrocytic differentiation of NSCs. However, human astrocyte function was impaired in the MECP2 silenced cells and these cells expressed lower levels of the glutamate transporter EAAT2. The signaling pathways involved in these effects of EPO are not fully characterized; however, NF-κB has been recently reported to regulate EAAT2 expression in astrocytes (*Wei et al., 2019*).

Finally, we found that both DMF and EPO rescued the decreased BDNF expression in MECP2-silenced microglia and astrocytes. Indeed, both NF-κB and NFE2L2 pathways have been reported to

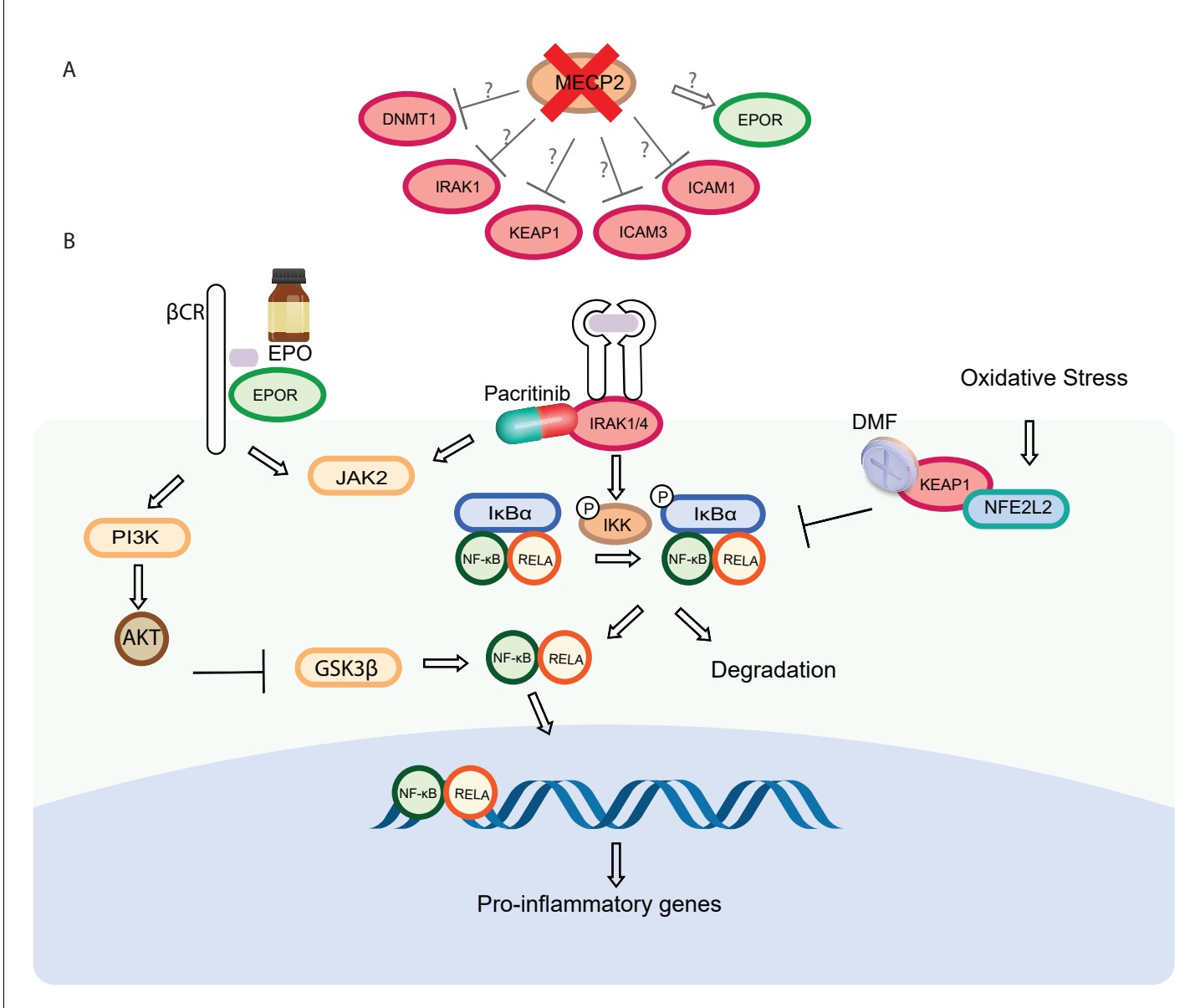

**Figure 7.** A model of MECP2 network genes converging on NF-κB signaling. (**A**) We propose that reduced MECP2 levels could lead to increase in ICAM1, ICAM3, IRAK1, and KEAP1 levels, and a decrease of EPOR levels within the tissue-protective receptor. (**B**) IRAK-1 plays a role in NF-κB activation through IKK activation, which leads to IκB phosphorylation and subsequent degradation, NF-κB nuclear translocation and the expression of its pro-inflammatory target genes. KEAP1 binds NFE2L2 and prevents its nuclear localization. NFE2L2 target genes inhibits NF-κB signaling through a non-transcriptional mechanism involving degradation of IκBα. EPO binds EPOR and enhances the activation of Akt, resulting in inhibition of GSK-3β and inhibition of NF-κB nuclear transport.

be associated with the regulation of BDNF expression (*Caviedes et al., 2017*; *Nair and Wong-Riley, 2016*). Thus, DMF can increase BDNF levels through an increase in NFE2L2, which binds the *BDNF* promotor and acts as a transcriptional activator (*Nair and Wong-Riley, 2016*). EPO administration activates the STAT3 signaling pathway (*Rey et al., 2019*), which can in turn elevate BDNF expression (*Chen et al., 2016*).

While the strong evolutionary linkage of our target proteins to MECP2 strongly suggest that they are mediating the reversal of MECP2 silencing phenotypes by their associated drugs, we have not ruled out the possibility for alternative action of these drugs or joint action upon our involving other protein targets. A good example of this are the HDAC chromatin modifiers, a group of genes which

did not come up as specifically linked to MECP2 in our evolutionary analysis but are well-known interactors. HDACs play a crucial role in neuronal development, and MECP2-mediated gene repression occurs mainly through HDAC recruitment via the Sin3a and NCoR complexes (*Lyst et al., 2013*; *Ebert et al., 2013*; *Nan et al., 1998*). HDACs have been suggested as therapeutic targets for Rett Syndrome (*Shukla and Tekwani, 2020*) and HDAC6 inhibition has been linked to improved BDNF trafficking in a Rett mouse model (*Xu et al., 2014*). Importantly, one of the compounds we examined, DMF, can reduce HDAC1/2/4 expression in rat astrocytes (*Kalinin et al., 2013*), and EPO can promote phosphorylation of HDAC5 in rat hippocampal neurons (*Jo et al., 2016*). It is thus plausible that HDAC inhibition mediates the effects we observe for these drugs, at least in part. However, the effects on HDAC activity are not the principal mechanisms of action of either of the proposed compounds (*Kalinin et al., 2013*; *Jo et al., 2016*), nor do they have established interactions with the HDAC most implicated in RTT biology (HDAC6). Future studies should investigate if any of the MECP2-rescuing activities of these drugs act through HDACs in concert with or independent of the target proteins identified here.

In summary, EPO, DMF and pacritinib act differentially on MECP2-silenced NSCs, microglia and astrocytes, providing a preliminary validation of our approach. The investigation into specific drug effects and mechanisms of actions are important areas for future studies, as are drug combination assays and in-vivo studies of EPO, DMF and pacritinib using MECP2 KO mice. These will be to assess whether any of the identified drugs can be further be explored for clinical use.

## Materials and methods

### Mapping MECP2 conservation along 1028 eukaryotic species and identifying correlated genes

We used the pipeline we described in *Tsaban et al., 2021* to calculate the Normalized Phylogenetic Profile (NPP) of 1028 eukaryotic species and rank genes based on their evolutionary similarity to MECP2. Briefly, 20,192 human proteins (one representative protein sequence per gene) were downloaded from UniProt reference proteomes (June 2018 release) (*Bateman and UniProt Consortium, 2019*). We used the Uniprot canonical isoform for each gene. For the 29 genes with more than one canonical isoform, we selected the longest isoform to maximize sequence information. The proteins were searched with BLASTP (*Camacho et al., 2009*) against the proteomes of the 1028 eukaryotes. A bit-score of 20.4, corresponding to a BLAST e-value of 0.05, was set as a minimal similarity threshold. The top scoring protein in each organism was selected. The bit-scores were normalized to protein length and phylogenetic distance from humans. The output is a matrix P of size 20,242 x 1028 where each entry Pab is the best BLASTP bit score between a human protein sequence 'a' and the top result in organism 'b'. The Pearson correlation was calculated for each profile with the MECP2 profile. The top 200 genes in all eukaryotes were selected as the E200 group for further analysis. This procedure was repeated using only the 51 mammal species, and the top 200 genes in were selected as the M200 group for further analysis.

### Filtering genes with drug interactions and constructing protein network

Drug gene interactions were collected from DGIdb (*Cotto et al., 2018*) and from Open Targets (*Oxford Academic, 2019*). Drugs at any stage of development were included in the search as long as the interaction with the target gene had known directionality (e.g. inhibitory, activating). Drug-gene interactions were validated through a literature review and known gene-gene interactions were collected from STRING (*Szklarczyk et al., 2019*). Network diagram was constructed using STRING and Cytoscape (*Lopes et al., 2011*).

### Synteny and co-localization analysis

Gene locations were retrieved from BioMart (*Kasprzyk, 2011*), the NCBI genome data viewer (*Figure 2—figure supplement 1A*, *Figure 2—figure supplement 2A*), and Genomicus (*Nguyen et al., 2018*; *Figure 3A*). Synteny in ancestral species was obtained from Genomicus (*Nguyen et al., 2018*). Intra-chromosomal Hi-C contact heatmaps and TAD borders were collected from the 3D Genome Browser (*Wang et al., 2018*), using GM12878 cell data from the high-resolution Hi-C dataset of *Rao et al., 2014*.

## Neural cell cultures

Immortalized human microglial cells and astrocytes were obtained from Applied Biological Material (Richmond, BC, Canada). Human NSCs (H9, hESC-derived) (ReNcell) were obtained from Invitrogen, Merck (Germany). Human neurons were obtained from ScienCell (Carlsbad, CA, USA). All cells employed in this study were tested for mycoplasma contamination (Mycoplasma PCR Detection Kit) and found negative.

## Transduction of neural cells

Lentivirus vectors (System Biosciences, Mountain View, CA, USA) expressing the MECP2 or control shRNAs and the reporters GFP/fLuc GFAP, mCherry/Luc MAP2 and NF-kB luciferase were packaged and used to transduce the cells according to the manufacturer's protocol and as previously described (*Giladi et al., 2015*).

## NSC differentiation

Human NSCs were maintained as spheroids in an NSC maintenance medium containing fibroblast growth factor 2 (FGF-2) and an epidermal growth factor (EGF, 20 ng/ml) on laminin-coated flasks. For differentiation, the cells were maintained in NSC maintenance medium without FGF-2 and EGF and neuronal and glial differentiation were observed.

## Microglia polarization

Human microglia cells were silenced for MECP2 using lentivirus vectors expressing MECP2 or control shRNAs. After 5e days, the expression of MECP2 was determined by western blot analysis and cells exhibiting a decrease of at least 80% were employed in further studies. Control and silenced cells were treated with the specific compounds and the expression of M1- and M2-associated markers were determined after 72 hr using RT-PCR. All experiments were done in triplicates and were repeated three times.

## Cytotoxicity assay

Cells were treated with different concentrations of DMF (1 10 and 50 mM), EPO (10, 30 and 50 ng/ml), and Pacritinib (1 and 10 mM). Following 3 days of treatment, the cells were analyzed for cell death using LDH assay. The results are presented as the means ± SD of six samples for each compound.

## Western blot analysis

Cell pellet preparation and Western blot analyses were performed as previously described (*Lomonaco et al., 2009*). Equal loading was verified using an anti-β-actin or tubulin antibodies as described (*Giladi et al., 2015*; *Bier et al., 2018*).

## Real-time PCR

Total RNA was extracted using RNeasy midi kit according to the manufacturer's instructions (Qiagen, Frederick, MD, USA). Reverse transcription reaction was carried out using 2 µg total RNA (*Bier et al., 2018*). The primer sequences are described in the supplementary files (*Supplementary file 3*).

## Phagocytosis analysis

Human microglial cells were silenced for MECP2. Phagocytosis was determined using the pHrodo Green zymosan bioparticle assay (Invitrogen, Carlsbad, CA, USA) according to the manufacturer's instructions. Briefly, microglia were incubated with a solution of pHrodo Green zymosan bioparticles in Live Cell Imaging Solution (0.5 mg/ml) for 2 hr. Phagocytosis was determined using a fluorescence plate reader at Ex/Em 509/533.

## Luciferase activity

The firefly luciferase activity of the NF-kB, GFAP and MAP two activities and the control Renilla luciferase activity were analyzed using the Dual-Luciferase Reporter Assay System (Promega Corporation).

### Statistical analysis for cell assays

The results are presented as the mean values ± SD. Data were analyzed using a Student's $t$-test with correction for data sets with unequal variances (Welch's unequal variances $t$-test).

## Acknowledgements

We thank Dr. Yaron Daniely and the Yissum team for being a major driving force for this project. We thank Dr. Liana Patt and the Integra Holdings team for their support. We thank Dr. Hae Kyung Lee and Susan Finniss for their technical support. Figures include vector images designed by Freepik. We thank Dr. Amir Eden and Prof. Yinon Ben Neriah for their insight and comments. Funding: This project was supported by an Israel Science Foundation grant 1591/19 to YT and by Integra Holdings. Additional support to BPB and IU came from a grant from the Beethoven Foundation.

## Additional information

### Funding

| Funder | Grant reference number | Author |
|---|---|---|
| Israel Science Foundation | 1591/19 | Yuval Tabach |
| Integra LifeSciences | | Yuval Tabach |
| Beethoven Foundation | | Benjamin P Berman |

The funders had no role in study design, data collection and interpretation, or the decision to submit the work for publication.

### Author contributions

Irene Unterman, Software, Formal analysis, Investigation, Visualization, Writing - original draft; Idit Bloch, Resources, Data curation, Software, Formal analysis; Simona Cazacu, Gila Kazimirsky, Validation, Investigation, Visualization, Methodology; Bruria Ben-Zeev, Conceptualization, Methodology, Writing - review and editing; Benjamin P Berman, Conceptualization, Resources, Supervision, Writing - original draft, Writing - review and editing; Chaya Brodie, Conceptualization, Supervision, Investigation, Methodology, Writing - original draft; Yuval Tabach, Conceptualization, Supervision, Funding acquisition, Writing - original draft, Writing - review and editing

### Author ORCIDs

Irene Unterman (iD) https://orcid.org/0000-0002-5697-9612
Simona Cazacu (iD) http://orcid.org/0000-0002-6085-4177
Benjamin P Berman (iD) http://orcid.org/0000-0002-2099-9005
Yuval Tabach (iD) https://orcid.org/0000-0001-9521-3217

### Decision letter and Author response

Decision letter https://doi.org/10.7554/eLife.67085.sa1
Author response https://doi.org/10.7554/eLife.67085.sa2

## Additional files

### Supplementary files

- Supplementary file 1. Genes most co-evolved with MECP2.
- Supplementary file 2. STRING GO-term enrichment of the E200 set.
- Supplementary file 3. RT-PCR primer sequences.
- Transparent reporting form

## Data availability

The phylogenetic data required to reproduce the analysis are openly available at Zenodo [https://zenodo.org/record/4464120#.YEoFpHmxVPY]. Code used for the analysis is openly available through Github https://github.com/Gulrene/MECP2_phylogeny (copy archived at https://archive.softwareheritage.org/swh:1:rev:14063b44a40688a8024a06347b63cfdac74b96ad).

The following previously published dataset was used:

| Author(s) | Year | Dataset title | Dataset URL | Database and Identifier |
|---|---|---|---|---|
| Stupp D, Tsaban T, Bloch I, Sharon E, Tabach Y | 2021 | CladeOScope: elucidating functional interactions via a clade co-evolution prism | https://zenodo.org/record/4464120#.YQjhGRMzael | Zenodo, 10.5281/zenodo.4464120 |

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
