## [Decision Letter]

**Acceptance summary:**

This manuscript will interest scientists who are aiming to leverage concepts and tools of evolutionary biology to identify novel gene targets, in this case towards much-needed therapeutic interventions for Rett syndrome. The follow up experiments presented in the paper are detailed, well thought out, and help establish the potential of the identified drugs in alleviating molecular signatures in in vitro disease models.

**Decision letter after peer review:**

Thank you for submitting your article "Expanding the MECP2 network using comparative genomics reveals potential therapeutic targets for Rett syndrome" for consideration by *eLife*. Your article has been reviewed by 2 peer reviewers, and the evaluation has been overseen by George Perry as the Senior and Reviewing Editor. The reviewers have opted to remain anonymous.

Essential revisions:

The manuscript has potential but requires major revisions for further consideration by *eLife*. In consultation, the editor and reviewers considered the level of revision to be possible, but we emphasize that we have high expectations, expecting at least the major comments made by each reviewer (below) to be robustly addressed. These essential revisions include improving the comprehensiveness of the comparative genomics pipeline and the need for more explicit validation experiments.

*Reviewer #1 (Recommendations for the authors):*

On line 367 – The differential mechanisms of restoration are interesting in the face of MeCP2s broad genomic role. While this is more of a future direction, could these treatments have a combinatorial beneficial effect? The in-vivo experiments that the authors are conducting should be very interesting. I wonder if all 33 identified genes could be screened for their toxicity/efficacy in these KD cells. The chosen 3 candidate proteins are all from the same evolutionarily conserved chromosomal cluster yet have meaningfully different cell specific mechanisms of RTT amelioration. Perhaps the other 33 identified genes with actionable druggable targets might have even more diverse mechanisms.

On line 51 there is a small typo, intro -> into.

*Reviewer #2 (Recommendations for the authors):*

1. Addressing/answering the following comments would help provide supporting evidence for the comparative evolutionary pipeline used by the authors:

a. Can the authors elaborate on their NPP methodology and how did they avoid the false positives?

b. Is the functional nature of genes different between E200 and M200?

c. Is the overlap of 10 genes between E200 and M200 statistically significant, given that overlapping input data was used? Is it higher or lower than expected?

d. Among the 390 genes, how many well established functional and biophysical partners of MECP2 were identified? What positive controls did the authors use to validate the robustness of their comparative genomic pipeline?

e. HDACs appear to be highly connected functional and biophysical partners of MECP2. Yet the authors do not seem to have identified any of them. Can the authors speculate on the reasons behind that?

f. How many novel MECP2 partners (without any previously known links) do the authors identify?

g. Some of the partners identified by the authors have functional association in Stringdb. Can the authors perform an enrichment analysis, with randomized controls, to demonstrate whether this functional enrichment is significantly more than expected? It will be an important validation for the authors PP approach.

2. The authors detect that 33 out of the 390 genes are druggable targets. Is that higher/lower than expected?

3. The authors show a series of sophisticated experiments to demonstrate the impact of the three drugs. The following clarifications will help strengthen the importance of the evolutionary pipeline to identify these drugs:

a. Are any of the previously known MECP2 functional partners targeted by EPO, DMF and Pacritinib in DGIdb and Open Targets? For example, a quick literature survey shows that all three molecules have direct or indirect effects on HDAC activity, which are functional and biophysical interaction partners of MECP2.

b. If the authors were to use known functional partners of MECP2 from Stringdb, PDB, or biogrid and check for their druggability in DGIdb and Open Targets, will the authors identify EPO, DMF and Pacritinib? If not, it proves that the comparative genomic pipeline was critical for identification of these drugs. If yes, it shows that the authors didn't need the pipeline. Or perhaps the authors mean to demonstrate that EPOR, KEAP1 and IRAK work upstream of HDACs and are therefore contributing to a better understanding of the mechanism of action of MECP2. If so, the authors do not have enough text, analysis or experiments to demonstrate that.

c. Can the authors perform silencing or knockdown experiment to demonstrate that the effects of the three drugs are mediated through EPOR, KEAP1 and IRAK and not through other targets?

---

## [Author Response]

Reviewer #1 (Recommendations for the authors):On line 367 – The differential mechanisms of restoration are interesting in the face of MeCP2s broad genomic role. While this is more of a future direction, could these treatments have a combinatorial beneficial effect? The in-vivo experiments that the authors are conducting should be very interesting. I wonder if all 33 identified genes could be screened for their toxicity/efficacy in these KD cells. The chosen 3 candidate proteins are all from the same evolutionarily conserved chromosomal cluster yet have meaningfully different cell specific mechanisms of RTT amelioration. Perhaps the other 33 identified genes with actionable druggable targets might have even more diverse mechanisms.

We tested the combinatorial effect of EPO with DMF on BDNF secretion, since these two both had an individual effect on this phenotype and Pacritinib did not (new Figure 6C). There may be an additive effect to EPO+DMF, compared to EPO or DMF alone, though we did not see a substantial synergistic effect. While the points you make are largely beyond the scope of this initial study, we did incorporate some of them in the improvements to our Discussion section discussing future directions (lines 420-423, 483).

We note that only 2 of the 3 candidate proteins are from the same chromosomal cluster (EPOR and KEAP1), while IRAK1 is from a conserved cluster with MECP2. We have modified the text on lines 202-204 to make this clearer.

On line 51 there is a small typo, intro → into.

Fixed.

Reviewer #2 (Recommendations for the authors):1. Addressing/answering the following comments would help provide supporting evidence for the comparative evolutionary pipeline used by the authors:a. Can the authors elaborate on their NPP methodology and how did they avoid the false positives?

We described our two new methodology and benchmarking studies in the new Introduction on lines 104-109. Bloch et al. 2020 explicitly deals with the specificity issue using control datasets from the CORUM and KEGG databases. We have found that the most important factor in reducing false positives is our incorporation of within-clade normalization (discussed in Bloch 2020). In addition to our standard approach, we tried to additionally limit false positives by selecting candidates that were top-ranked in both the mammalian and eukaryotic lists, and that clustered within an evolutionarily conserved and topologically organized chromosomal domain.

b. Is the functional nature of genes different between E200 and M200?

We have done a new analysis of Gene Ontology and other functional annotations using STRING (now added as supplementary file 2). The E200 list is enriched in a number of different functional categories, with some of the most significant related to innate immunity and the immune response. These included MHC Class II receptor activity and peptidoglycan receptor activity. Consistent with this, autoimmunity annotations such as Asthma, Allograft rejection, and Type I diabetes are highly enriched in the KEGG network. In contrast, the M200 list has almost no functional enrichment, except for the general “Disease” category within Uniprot annotated keywords. This indicates that the two lists might be revealing fundamentally different relationships to MECP2, and makes the 10 overlapping genes especially interesting. We have added a description of this to the Results section in lines 152-159.

c. Is the overlap of 10 genes between E200 and M200 statistically significant, given that overlapping input data was used? Is it higher or lower than expected?

It is statistically significant if you consider the two lists independent (p<3.2E-4 by the hypergeometric test), but as you point out the phylogenetic correlation in Eukaryotes can be affected by the correlation within the mammalian clade. We do not know a way to factor this out from the significance estimate, so we have just edited the description in the text to reflect this fact (lines 160-165).

d. Among the 390 genes, how many well established functional and biophysical partners of MECP2 were identified? What positive controls did the authors use to validate the robustness of their comparative genomic pipeline?

In the original manuscript, we relied on the fact that the 10 proteins in the E200/M200 intersection list included the well-established MECP2-interacting protein IRAK1. Given that evidence, and the fact that we have used this pipeline before and validated it extensively (Bloch et al., 2020; Tsaban et al., 2021), this gave us enough confidence to proceed to functional drug follow-up. However, based on your feedback we have now performed new comparisons to STRING to give a more comprehensive analysis for MECP2. See our response to question 1g, below.

e. HDACs appear to be highly connected functional and biophysical partners of MECP2. Yet the authors do not seem to have identified any of them. Can the authors speculate on the reasons behind that?

First, we investigated only the very most strongly linked of our PP lists, which would necessarily leave out some true interactions. However, we have gone back to our full original lists and found that the relevant HDACs have quite low PP similarity scores to MECP2 in general. We have plotted the phylogenetic profiles for MECP and the two major classes of HDACs in Author response image 1, showing significant differences. MECP2 is distinguished by a relative loss among a large group of vertebrates, and stronger conservation among mammals. Class I HDACS 1/2/3/8 clustered together and were much better conserved in ancient branches. Class II HDACS 4/5/6/7/9/10 clustered together and were characterized by loss within a sub-branch of fungi and all non-fungi single-cell eukaryotes (“other organisms”). While it is hard to interpret these differences at more ancient time points, the split within the non-mammalian vertebrates may be important in understanding the divergence between MECP2 and HDACs. While we did not add this more in depth analysis of HDAC evolutionary conservation, we have added new text describing the HDAC role in MECP2 function and RTT in our Introduction on lines 72-80 and their potential role as a mediator of our drug candidates in the Discussion on lines 462-479.

**Author response image 1. sa2fig1:** Normalized phylogenetic profiles of MECP2 and HDAC proteins.

f. How many novel MECP2 partners (without any previously known links) do the authors identify?

As shown in Figure 1—figure supplement 1D), there are 1,398 known interactions with MECP2 in the String-db categories “text-mining”, “co-expression”, and “experimental”. 366 of 390 genes in out E200+M200 list were not among these genes, indicating that large majority of our genes are novel relationships. We have added text describing this on lines 168-171.

g. Some of the partners identified by the authors have functional association in Stringdb. Can the authors perform an enrichment analysis, with randomized controls, to demonstrate whether this functional enrichment is significantly more than expected? It will be an important validation for the authors PP approach.

As described in response to your question 1f, only 24 of 390 genes in our E200+M200 gene lists (6.2%) had Stringdb associations with MECP2. Given that 1,398 (7.3%) of 19,257 total genes in Stringdb are associated with MECP2, this is compatible with random chance. However, in the 10 high priority genes included in both the M200 and E200 groups, 3 overlapped Stringdb (Author response image 2). The chance of observing 3 or more overlaps is 0.03 by the hypergeometric test. While this is not a high degree of overlap, it does help to establish the validity of our approach, along with our previous benchmarking of our pipeline using the CORUM, REACTOME, and KEGG databases (Bloch et al., 2020; Tsaban et al., 2021). These recent benchmarking studies are now described in lines 104-109. We also analyzed the 390 gene list using GeneAnalytics, and they were most enriched with genes expressed in the brain compared to all other tissues (see Figure 1—figure supplement 1C ).

**Author response image 2. sa2fig2:** Overlap between the E200 and M200 intersection set (PP) and all STRING MECP2 interactions.

2. The authors detect that 33 out of the 390 genes are druggable targets. Is that higher/lower than expected?

We performed random sampling of 390 genes from our PP input gene list. This resulted in an average of 39.9 druggable genes (std 6.0). Thus our gene lists was not significantly either enriched or depleted in druggable genes. This does not seem surprising to us, given that druggability depends on priorities of past research efforts and chemical properties of the gene product, rather than anything having to do with evolutionary links. We have added a sentence describing this to the Results section (line 191). If we missed your point here, please let us know.

3. The authors show a series of sophisticated experiments to demonstrate the impact of the three drugs. The following clarifications will help strengthen the importance of the evolutionary pipeline to identify these drugs:a. Are any of the previously known MECP2 functional partners targeted by EPO, DMF and Pacritinib in DGIdb and Open Targets? For example, a quick literature survey shows that all three molecules have direct or indirect effects on HDAC activity, which are functional and biophysical interaction partners of MECP2.

We addressed HDACs as well as other known associations with MECP2 , by performing an analysis of the top linked MECP2 genes from the STRING database. Neither HDACs nor other highly-ranked MECP2 associated genes were linked to either EPO or DMF. We used STRING as our baseline method for finding known interactions, since it includes text-mining, co-expression, and experimental associations (which includes links between MECP2 and many HDACs). Additionally, we searched for direct HDAC interactions for all class 1 and 2 HDACs in DGIDB, GeneCards and OpenTargets, and found no links to the 3 drugs we identified here. Below is a list of 46 drugs with direct HDAC association we found through this process.

**Table resptable1:** 

Gene	Drug	Interaction	Gene	Drug	Interaction
HDAC1	TACEDINALINE	inhibitor	HDAC 1	VORINOSTA T	inhibitor
HDAC1	PANOBINOSTAT LACTATE	inhibitor	HDAC4	TASQUINIMOD	allosteric modulator
HDAC1	ROMIDEPSIN	antagonist|inhibitor	HDAC6	BUFEXAMAC	inhibitor
HDAC1	FIMEPINOSTAT	inhibitor	HDAC6	TUBACIN	inhibitor
HDAC1	CUDC-101	inhibitor	HDAC1	Valproic acid	inhibitor
HDAC1	SCRIPTAID	inhibitor	HDAC1	Fingolimod	inhibitor
HDAC1	RESMINOSTAT	inhibitor	HDAC1	4SC-202	inhibitor
HDAC1	NANATINOSTAT	inhibitor	HDAC1	sodium phenylbutyrate	inhibitor
HDAC1	APICIDIN	inhibitor	HDAC1	CHR-3996	inhibitor
HDAC1	ABEXINOSTAT	inhibitor	HDAC1	MGCD-0103	inhibitor
HDAC1	BELINOSTAT	inhibitor	HDAC1	PCI-24781	inhibitor
HDAC1	DEPAKOTE	inhibitor	HDAC1	Pivanex	inhibitor
HDAC1	MOCETINOSTAT	inhibitor	HDAC1	SB939	inhibitor
HDAC1	DACINOSTAT	inhibitor	HDAC2	Theophylline	activator
HDAC1	PANOBINOSTAT	inhibitor	HDAC2	Aminophylline	activator
HDAC1	AN-9	inhibitor	HDAC2	Atorvastatin	inhibitor
HDAC1	TUCIDINOSTAT	inhibitor	HDAC2	Fluvastatin	inhibitor
HDAC1	RICOLINOSTAT	inhibitor	HDAC2	Oxtriphylline	activator
HDAC1	ENTINOSTAT	inhibitor	HDAC2	Pravastatin	inhibitor
HDAC1	GIVINOSTAT	inhibitor	HDAC2	Simvastatin	inhibitor
HDAC1	CITARINOSTAT	inhibitor	HDAC3	Romidepsin	inhibitor
HDAC1	QUISINOSTAT	inhibitor	HDAC3	RG2833	inhibitor
HDAC1	PRACINOSTAT	inhibitor	HDAC7	TACEDINALINE	inhibitor

b. If the authors were to use known functional partners of MECP2 from Stringdb, PDB, or biogrid and check for their druggability in DGIdb and Open Targets, will the authors identify EPO, DMF and Pacritinib? If not, it proves that the comparative genomic pipeline was critical for identification of these drugs. If yes, it shows that the authors didn't need the pipeline. Or perhaps the authors mean to demonstrate that EPOR, KEAP1 and IRAK work upstream of HDACs and are therefore contributing to a better understanding of the mechanism of action of MECP2. If so, the authors do not have enough text, analysis or experiments to demonstrate that.

For STRING, please see our response to question 3a above. Biogrid only contains only 182 proteins linked to MECP2, and none are linked to the 3 drugs tested here through these databases. This reinforces what we have shown previously, that PP is complementary to other functional relationship databases. We were not sure what you had in mind for PDB. Only one protein (TBLR1) has a PDB structure in complex with MECP2, and this gene is not linked to any of our 3 drugs. There are three structures with DMF in complex with a human gene (SMN1, UHRF1, TDRD3), two with Pacritinib (NQO2, SMYD3), and none with EPO. Only UHRF1 has any connection to MECP2 through STRING.

c. Can the authors perform silencing or knockdown experiment to demonstrate that the effects of the three drugs are mediated through EPOR, KEAP1 and IRAK and not through other targets?

This is a completely valid point and definitely an important area for future work, as evidenced by the point you brought up about HDACs. We have now added text to our Discussion regarding this on lines 478-483. Given our resources, these experiments would be beyond the scope of what we could accomplish in a short time frame. We believe that our results here, while not 100% conclusive, are highly compelling given the fact that all three drugs predicted by our pipeline showed reversal of multiple Rett-like phenotypes of MECP2 depletion. It seems highly unlikely that by chance we picked three drugs that could reverse these changes in the MECP2 depleted condition but have no effect on the wildtype cells. We believe it is important to publish our initial findings based on this methodological workflow, as well as our gene and drug lists, so that other labs may follow up on these results in parallel to us.

References:

Bateman, A. (2019). UniProt: A worldwide hub of protein knowledge. Nucleic Acids Research, 47(D1), D506–D515. https://doi.org/10.1093/nar/gky1049

Bloch, I., Sherill-Rofe, D., Stupp, D., Unterman, I., Beer, H., Sharon, E., and Tabach, Y. (2020). Optimization of co-evolution analysis through phylogenetic profiling reveals pathway-specific signals. Bioinformatics, 36(14), 4116–4125. https://doi.org/10.1093/bioinformatics/btaa281

Dunwell, T. L., Paps, J., and Holland, P. W. H. (2017). Novel and divergent genes in the evolution of placental mammals. Proceedings of the Royal Society B: Biological Sciences, 284(1864). https://doi.org/10.1098/rspb.2017.1357

Fuchs, S. B. A., Lieder, I., Stelzer, G., Mazor, Y., Buzhor, E., Kaplan, S., … Shtrichman, R. (2016). GeneAnalytics: An Integrative Gene Set Analysis Tool for Next Generation Sequencing, RNAseq and Microarray Data. OMICS A Journal of Integrative Biology, 20(3), 139–151. https://doi.org/10.1089/omi.2015.0168

Philippon, H., Souvane, A., Brochier-Armanet, C., and Perrière, G. (2017). IsoSel: Protein isoform selector for phylogenetic reconstructions. PLoS ONE, 12(3).

https://doi.org/10.1371/journal.pone.0174250

Tsaban, T., Stupp, D., Sherill-Rofe, D., Bloch, I., Sharon, E., Schueler-Furman, O., … Tabach, Y. (2021).

CladeOScope: functional interactions through the prism of clade-wise co-evolution. NAR Genomics and Bioinformatics, 3(2). https://doi.org/10.1093/nargab/lqab024

Villanueva-Cañas, J. L., Laurie, S., and Albà, M. M. (2013). Improving genome-wide scans of positive selection by using protein isoforms of similar length. Genome Biology and Evolution, 5(2), 457– 467. https://doi.org/10.1093/gbe/evt017